# Meta-learning local learning rules for structured credit assignment with sparse feedback

## Abstract

Biological neural networks learn complex behaviors from sparse, delayed feedback using local synaptic plasticity, yet the mechanisms enabling structured credit assignment remain elusive. In contrast, artificial recurrent networks solving similar tasks typically rely on biologically implausible global learning rules or hand-crafted local updates. The space of local plasticity rules capable of supporting learning from delayed reinforcement remains largely unexplored. Here, we present a meta-learning framework that discovers local learning rules for structured credit assignment in recurrent networks trained with sparse feedback. Our approach interleaves local neo-Hebbian-like updates during task execution with an outer loop that optimizes plasticity parameters via **backpropagation through learning (BPTL)**. The resulting three-factor learning rules enable long-timescale credit assignment using only local information and delayed rewards, offering new insights into biologically grounded mechanisms for learning in recurrent circuits.

---

## 1 Introduction

Learning in biological organisms involves changes in synaptic connections (synaptic plasticity) between neurons [1, 2]. Synaptic changes are believed to underlie memory formation and are essential for adaptive behaviour [3]. Experimental evidence suggests that synaptic changes depend on the co-activation of pre- and postsynaptic activity [4, 5], and possibly other local variables available at the synaptic site [6, 7]. These unsupervised synaptic modifications have explained activity-dependent circuit refinement during development such as the emergence of functional properties like receptive field formation based on naturalistic input statistics [8].

Yet, most organisms routinely solve complex tasks that require feedback through explicit supervisory or reinforcement signals. These signals are believed to gate or modulate plasticity, acting in the form of a third factor that scales and also probably imposes the direction of the synaptic modifications [9]. How error- or reward-related information is propagated through the recurrent interactions is not yet clear. While prior work has largely focused on hand-crafted synaptic updates for unsupervised self-organization, or biologically plausible approximations of backpropagation [10], the space of plasticity rules capable of supporting structured credit assignment from delayed feedback remains vastly underexplored.

Backpropagation through time (BPTT), the standard approach for training recurrent neural networks (RNNs), is biologically implausible since it requires symmetric forward and backward connections and non-local information [11, 12]. Although recent work has reformulated BPTT into more biologically plausible variants using random feedback [13], truncated approximations [14], or by learning

feedback pathways [15], these methods require continuous error signals to refine recurrent connections.

Here, we adopt a bottom-up approach: instead of imposing hand-designed synaptic rules, we discover biologically plausible plasticity rules that support learning through delayed reinforcement signals via meta-optimisation. Building on recent work [16], we parameterise plasticity rules as functions of local signals (presynaptic activity, postsynaptic activity, and synapse size) and meta-learn their parameters within a second reinforcement learning loop. With that, our present work tackles the following questions:

- **Which local learning rules can implement structured credit assignment under biological constraints?**
- **Do different forms of plasticity give rise to different computational regimes and representations as observed with gradient based training (e.g., "lazy" vs. "rich" learning)?**

Recent theory distinguishes between lazy and rich regimes of learning in RNNs: in the lazy regime, representations remain fixed while output weights adapt; in the rich regime, the network reorganises its internal dynamics to encode task structure. While these regimes are well-characterised for gradient-trained networks, it remains unclear whether biologically plausible learning rules can support either or both, and what synaptic mechanisms underlie each regime. Here we demonstrate that different forms of plasticity naturally lead to qualitatively different learning trajectories and internal representations, akin to their gradient-based learning rules.

## 2 Method

**Network dynamics.** We consider recurrent neural networks (RNNs) of firing rate neurons coupled through a synaptic matrix $W \in \mathbb{R}^{N \times N}$ [17], with additional input and output matrices $W_{\text{in}} \in \mathbb{R}^{N_{\text{in}} \times N}$ and $W_{\text{out}} \in \mathbb{R}^{N \times N_{\text{out}}}$ that route task-relevant input into the recurrent circuit and read out network activation to generate task-specific outputs (actions). The equations governing the network dynamics are

$$\frac{\mathrm{d}\mathbf{x}^t}{\mathrm{d}t} = -\mathbf{x}^t + W\boldsymbol{\phi}(\mathbf{x}^t) + W_{\text{in}}\mathbf{u}^t, \tag{1}$$

$$\mathbf{r}^t = \boldsymbol{\phi}(\mathbf{x}^t) \doteq \tanh(\mathbf{x}^t), \tag{2}$$

where $\mathbf{x}^t \in \mathbb{R}^N$ is the vector of pre-activations (or input currents) to each neuron in the network, $\boldsymbol{\phi}(\cdot) : \mathbb{R}^N \to \mathbb{R}^N$ denotes the single-neuron transfer functions, $\mathbf{r}^t \in \mathbb{R}^N_+$ is the vector of instantaneous firing rates, $\mathbf{u}^t$ stands for the activity of the $N_{\text{in}}$ input neurons. In the terms above, the $\cdot^t$ superscript indicates time dependence. Network outputs $\mathbf{z}^t$ are obtained from linear read-out neurons as

$$\mathbf{z}^t = W_{\text{out}}\mathbf{r}^t. \tag{3}$$

**Sparse feedback and parametrized learning rules.** We consider networks that learn context-dependent cognitive tasks using biologically plausible local learning rules, guided by sparse reinforcement signals $R$ provided only at the end of each training episode. To enable learning from such delayed and global signals, each synapse between a pre-synaptic unit $j$ and a post-synaptic unit $i$ maintains an eligibility trace $e_{ij}$ [18], which integrates the history of (co-)activation during the episode. We define the evolution of eligibility traces with differential equations of the form

$$\frac{\mathrm{d}e_{ij}^t}{\mathrm{d}t} = \mathcal{H}_\theta(r_j^t, x_i^t) - \frac{e_{ij}}{\tau_e} = \sum_{0 \le k;l \le d;} \theta_{k,l} \left(r_j^t\right)^k \left(\bar{x}_i - x_i^t\right)^l - \frac{e_{ij}}{\tau_e}, \tag{4}$$

where $\tau_e$ is a decay time-scale, $\bar{x}_i$ is a running average of the pre-activation of neuron $i$, and $\theta_{k,l} \in \mathbb{R}$ are learnable coefficients. In contrast to eligibility traces based solely on pairwise correlations [19], we use here a polynomial expression that captures richer interactions between pre- and post-synaptic activity. Each coefficient $\theta_{k,l}$ can be construed as a term-specific learning rate, which may be positive (Hebbian), negative (anti-Hebbian). This parameterization allows individual terms to modulate synaptic eligibility based on pre-synaptic activity, post-synaptic activity, co-activity, or deviations from a homeostatic set point. In our experiments, we set $d = 2$, yielding 9 monomial terms that capture nonlinearities and interaction effects, while remaining computationally tractable.

The recurrent weight matrix $W$ gets updated at the end of each training episode according to a reward-modulated learning rule

$$\Delta w_{ij} = e_{ij}\left(R - \bar{R}\right) - \frac{w_{ij}}{\tau_w}, \tag{5}$$

where $\tau_w$ denotes the time scale of weight decay, $e_{ij}$ stands for the eligibility trace accumulated during the episode, while $R$, $\bar{R}$ stand for the obtained and the expected reward. Here, we model reward expectations for each type of trial independently as a running average of past rewards for this trial type [20]. This update rule enables credit assignment through the interaction between synaptic eligibility and trial-specific reward prediction error, consistent with neo-Hebbian three-factor learning rules hypothesized to operate in biological circuits [19]. In principle the weight updates happen due to (slow) weight decay or due to reward prediction errors.

**Meta-learning plasticity rules.** While previous work has relied on hand-crafted eligibility trace dynamics and synaptic update rules to train recurrent neural networks with sparse feedback [20], we instead adopt a meta-learning approach to learn the parameters of the plasticity rules. Our framework consists of two nested training loops: **(i)** an inner loop in which the recurrent network is trained over several episodes using local learning rules and sparse reinforcement signals provided at the end of each episode, as described above; and **(ii)** an outer loop that optimizes the plasticity meta-parameters $\Theta = \{\{\theta_{k,l}\}_{k,l=0}^2, \tau_w, \tau_e\}$ via gradient descent using **backpropagation through learning** on a meta-loss computed over $K$ training episodes (trials). This approach allows the learning rules themselves to be adapted to the task, rather than be fixed a priori.

**Backpropagation through learning.** Our goal is to optimise the learning rule parameters $\theta$ to maximise task performance, measured as the expected cumulative reward $\langle R \rangle$ obtained after a fixed number of learning episodes. However, the reward $R$ obtained by the agent depends on the network's output, which in turn is determined by its synaptic weights $\mathcal{W} = \{W_{in}, W, W_{out}\}$. The weights are dynamically updated according to the employed synaptic update rule (Eq. 5). This plasticity rule, depends on the eligibility traces $e_{ij}$, which themselves are parameterised by $\theta$. This establishes a complex dependency chain over the network parameters: $R \leftarrow W \leftarrow e \leftarrow \theta$. Thus directly computing the gradient $\nabla_\theta \langle R \rangle$ by backpropagating through the entire network dynamics over learning is computationally challenging.

To address this, we employ a REINFORCE-inspired approximation [21] to estimate the gradient $\nabla_\theta \langle R \rangle$. Recall that the REINFORCE gradient formula involves computing the gradient of an expected value by observing outcomes and scaling a measure of what elicited that outcome with the associated reward. Or more formally, scaling the gradient of the log-probability of an outcome with the reward associated with that outcome

$$\nabla_\theta \langle R \rangle = \langle (R - \bar{R}) \cdot \nabla_\theta \log \pi(R \mid \theta) \rangle \tag{6}$$

Here, since we consider deterministic weight updates, we do not have a stochastic policy $\pi$, as is common in policy gradient methods in reinforcement learning. However, we can consider the final weight configuration $\mathcal{W}(\Theta)$ as an *implicit policy* with parameters $\Theta$, that determine the learned network behaviour. We then use the **reward prediction error**, defined as $\delta R = R - \bar{R}$ (where $\bar{R}$ is a running average of the reward), as a signal to adapt the parameters $\theta$

$$\nabla_\theta \langle R \rangle \approx (R - \bar{R}) \cdot \frac{dW}{d\theta}. \tag{7}$$

Since the weight updates depend linearly on the eligibility trace (Eq. 5), we have

$$\frac{dW_{ij}}{d\theta_{kl}} = \delta R \cdot \frac{de_{ij}}{d\theta_{kl}}. \tag{8}$$

To relate this to the gradient of the reward with respect to $\theta$, we sum over all synapses, resulting in the approximation

$$\nabla_\theta \langle R \rangle \approx \sum_{i,j} \delta R \cdot \frac{de_{ij}}{d\theta_{kl}} = \sum_{i,j} \delta R \cdot \left(r_j^t\right)^k \left(\bar{x}_i^t - x_i^t\right)^l. \tag{9}$$

The eligibility trace $e_{ij}$ is a function of neural activity, and its dependency on the parameters $\theta$ is explicitly defined by the model (Eq. 4). For the eligibility trace parametrised in the polynomial

form of Eq. 4, the term $\frac{de_{ij}}{d\theta}$ has an explicit expression in terms of neural activations and firing rates (Eq. 9). This expression is fully analytic and requires no gradient propagation through the network or the learning episodes. The plasticity parameters $\theta$ are then updated using gradient ascent based on this estimated gradient.

To enforce sparsity on the identified rules in order to minimise the number of active terms in the identified rule to render it interpretable.

# 3 Results

We defer the reader to the Extensive results section in the Supplementary Information for the results of the numerical experiments.

# 4 Related work

Decades of research on synaptic plasticity have focused on hand-crafted learning rules designed to replicate experimentally observed changes in post-synaptic potentials from single-neuron recordings. However, the recent explosion in large-scale functional recordings, particularly longitudinal data collected across learning, has sparked growing interest in identifying the types of plasticity rules that may underlie observed changes in neural activity and behavioural performance. Despite this interest, the task remains extremely challenging: current experimental techniques do not allow direct measurement of synaptic interactions across large neural populations, making it difficult to infer the underlying synaptic mechanisms at play. Thus an increasing number of frameworks have emerged that aim to discover plasticity rules from indirect signatures such as changes in neural activity distributions, recorded trajectories, or behavioural performance. These approaches differ in what kind of observations they use, and in the assumptions they make about the network structure, plasticity rule parameterisation, and underlying task.

**Matching rate distributions.** One line of work focuses on inferring synaptic plasticity rules from pre- and post-learning firing rate distributions. Lim et al.[22] jointly infer neuron transfer functions and synaptic updates from observed rate distributions, under assumptions of Poisson firing statistics and linearized plasticity. This approach was later extended using Gaussian process priors over plasticity functions[23], improving flexibility but still restricted to feedforward networks and ignoring temporal dynamics.

These approaches do not model the full trajectory of activity during learning, instead identify plasticity rules that explain cumulative changes across learning. As a result, they cannot constrain rule parameters based on how learning unfolded in time.

**Inference by conditioning on neural trajectories.** A second group of methods exploits neural activity trajectories recorded over learning. Ramesh et al. [24] use a generative adversarial framework to infer plasticity rules that generate neural trajectories similar to empirical ones. While highly expressive, this method requires extensive data and computational resources, and suffers from known instability issues in GAN training. Confavreux et al. [16] proposed a meta-learning framework to discover plasticity rules that produce desired temporal coding properties in rate-based networks. While insightful, their approach optimises for a fixed synthetic objective (e.g., encoding elapsed time), rather than learning from observed data or behaviour.

**Behavior-based plasticity inference.** A third set of studies use behavioural performance trajectories to constrain synaptic plasticity. Ashwood et al.[25] fit learning rule parameters in rodent decision tasks using a Bayesian model, requiring approximation of the full posterior over synaptic weights. Rajagopalan et al.[26] reformulate the plasticity inference problem as logistic regression by assuming presynaptic activity and reward as the only inputs. These frameworks remain limited in flexibility, often neglecting dependencies on postsynaptic activity or synapse strength, which are essential for biologically grounded learning.

Most of these approaches assume feed-forward structure of the underlying network [23, 27], and consider plasticity evolving network dynamics in an unsupervised setting. Only the recent work of

169 [27] considers a reward term in the plasticity rule, that effectively puts the learning framework under
170 a reinforcement learning and thus closer to how biological organisms learn.

## 5 Limitations

172 Despite its strengths, our work has several limitations that point to opportunities for future improve-
173 ment and extension. One limitation is that the proposed meta-learning procedure must be run mul-
174 tiple times independently to discover multiple plasticity rules that satisfy the same task constraints.
175 Recent advances using simulation-based inference [16] provide a promising alternative for sampling
176 entire distributions over plasticity rules that solve a given cognitive task, potentially offering a more
177 efficient and principled exploration of solution space. Yet, simulation based inference is easy to
178 incorporate in our setting.

179 Another limitation is that our current framework is purely exploratory and does not explicitly in-
180 corporate constraints from experimentally recorded neural activity. While this allows for a broad
181 and flexible search over possible learning mechanisms, it limits the biological specificity of the dis-
182 covered rules. Extending our framework to incorporate such constraints, for instance, by biasing
183 the meta-optimisation toward activity trajectories consistent with recorded data, could yield more
184 realistic models of synaptic updates.

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
