# OpenReview forum: "Meta-learning local learning rules for structured credit assignment with sparse feedback"
_NeurIPS.cc/2025/Conference — Submitted to NeurIPS 2025_

### Official Review · Reviewer_pf5X · 2025-06-21

**Clarity:** 1
**Significance:** 1
**Originality:** 1
**Rating:** 1
**Confidence:** 3

**Summary:**

This is an incomplete submission missing experiments verifying the main claims of the paper, and therefore cannot be properly reviewed. The authors mention "We defer the reader to the Extensive results section in the Supplementary Information for the results of the numerical experiments", however the supplementary materials does not contain such result section.

**Questions:**

This is an incomplete submission missing experiments verifying the main claims of the paper, and therefore cannot be properly reviewed. The authors mention "We defer the reader to the Extensive results section in the Supplementary Information for the results of the numerical experiments", however the supplementary materials does not contain such result section.

**Ethical Concerns:**

["NO or VERY MINOR ethics concerns only"]

**Final Justification:**

As the paper does not contain any results section, I stand by my decision for rejection.

**Limitations:**

/

**Paper Formatting Concerns:**

/

**Quality:**

1

**Strengths And Weaknesses:**

This is an incomplete submission missing experiments verifying the main claims of the paper, and therefore cannot be properly reviewed. The authors mention "We defer the reader to the Extensive results section in the Supplementary Information for the results of the numerical experiments", however the supplementary materials does not contain such result section.

---

> ### Author Rebuttal · Authors · 2025-07-30
>
> Thank you very much for your time and effort spent to review our paper. Yes indeed at the time of the submission we didn't have the time to include trustworthy results, but he hoped that we would be able to upload a pdf during the revisions. It looks that this is not possible this time.

---

> > ### Comment · Reviewer_pf5X · 2025-08-05
> >
> > Thank you for being open about this. As the paper does not contain any results, I stand by my recommendation for rejection.

---

### Official Review · Reviewer_XwW4 · 2025-06-25

**Clarity:** 2
**Significance:** 2
**Originality:** 1
**Rating:** 2
**Confidence:** 4

**Summary:**

The paper proposes a meta-learning approach (i.e. an outer training loop) to learn local rules controlling the weights between the neurons of a RNN. The paper discuss how it aims at looking at how internal representations are affected by different local learning rules.

**Questions:**

- Results are missing, both regarding the performance of the algorithm solving tasks and how the internal representations are affected by different rules.

**Ethical Concerns:**

["NO or VERY MINOR ethics concerns only"]

**Final Justification:**

Since the paper doesn't provide results.. it's a clear reject.

**Limitations:**

yes

**Paper Formatting Concerns:**

Results are not included.

**Quality:**

2

**Strengths And Weaknesses:**

## Strengths

- The paper poses an interesting problem: learning local synaptic rules in ANN and investigate how different rules result in different internal representations in the RNN.


## Weakness

- No results of using the proposed algorithm to solve any task seem to have been included—neither in the main text nor in the appendix. Section 3 reads ““We defer the reader to the Extensive results section” ([pdf](zotero://open-pdf/library/items/LDSDZWNH?page=4)) but there isn’t any results section in the appendix.

- The method is very similar to existing meta-learning algorithms to learn local rules, e.g.,:

    - Rosenfeld B, Rajendran B, Simeone O. Fast on-device adaptation for spiking neural networks via online-within-online meta-learning. In2021 IEEE Data Science and Learning Workshop (DSLW) 2021 Jun 5 (pp. 1-6). IEEE.

    - Najarro E, Risi S. Meta-learning through hebbian plasticity in random networks. Advances in Neural Information Processing Systems. 2020;33:20719-31.

    - Stewart KM, Neftci EO. Meta-learning spiking neural networks with surrogate gradient descent. Neuromorphic Computing and Engineering. 2022 Sep 30;2(4):044002.


- Paper states: “While previous work has relied on hand-crafted eligibility trace  dynamics and synaptic update rules to train recurrent neural networks with sparse feedback, we instead adopt a meta-learning approach to learn the parameters of the plasticity rules.” Examples of previous work that do the also meta-learn local rules:

    - Pedersen JW, Risi S. Evolving and merging hebbian learning rules: increasing generalization by decreasing the number of rules. In Proceedings of the Genetic and Evolutionary Computation Conference 2021 Jun 26 (pp. 892-900).

- The paper present the learning approach as contrasting to ANN which “rely on biologically implausible global learning rules”; and “using only local information”. However, the proposed method  keeps a history of the trace and co-activations between the neurons. It’s not clear how the trace and the co-activation history are stored in differential equation (4) such that information remains purely local.

- Offloading the whole results sections to the appendix seems like bending the spirit of having an appendix a bit too much as it provides an unfair space advantage to the authors wrt to other submissions. According to the guidelines, appendix is meant to support the results presented in the paper.

- Some relevant work not discussed:

    - Tyulmankov D, Yang GR, Abbott LF. Meta-learning synaptic plasticity and memory addressing for continual familiarity detection. Neuron. 2022 Feb 2;110(3):544-57.

    - See references in here for more missing citations: Schmidgall S, Ziaei R, Achterberg J, Kirsch L, Hajiseyedrazi S, Eshraghian J. Brain-inspired learning in artificial neural networks: a review. APL Machine Learning. 2024 Jun 1;2(2).

---

> ### Author Rebuttal · Authors · 2025-07-30
>
> We are grateful for your feedback and the extensive reference list you provided. We are grateful that you find the problem interesting as we do. We were not aware of a subset of the papers you mention, so we are sure your proposal will improve the next iteration of our paper.  Thank you for taking the time to review it.

---

### Official Review · Reviewer_3MaT · 2025-06-30

**Clarity:** 1
**Significance:** 1
**Originality:** 1
**Rating:** 1
**Confidence:** 5

**Summary:**

The authors want to learn parameters of three-factor rules with eligibility traces using backpropagation. They parameterize the eligibility traces with a (polynomial of postsynaptic firing rates) times (presynaptic rate minus average) and a decay tau_e . This is a valid way to write an eligibility trace and overall the approach has a valid aim. Unfortunately, the result section is absent in the main text. The main text refers to supplementary.  The supplement file that I could download starts at page 16 and extends the theory, but also does not contain any results.

**Questions:**

comments:
1. tau_e is the time scale of the eligibility trace. It  should be around 1 second according to experimental data . In Eq. (24) the decay is replaced by a rectangular window corresponding to tau_e to infinity. This replacement is ad hoc and not mentioned in the main text.
Note that with REINFORCE (as opposed to TD learning) it is close to impossible to learn tasks with rewards that are delayed longer than 2 tau_e (whereas it is simple to learn with TD!).

2.  Eq. (6-9) in main text: I find that the detour to log-policy  in Eq. (6) is not helpful (it is not even a useful analogy).
Moreover Eq (7) does not lead to Eq. (8).   Eq. (8) is correct, but combined with Eq. (7) it does not yield Eq. 9).  Equation (19) has a square and  is different from Eq. (9).  Probably Eq. (9) has a mistake, because Figure 1 in Supplementary and Eq. (24) give a different equation.

3. Could the authors start with a Loss (R - bar{R})^2. ? Equation (7) is then found using chain rule if the term dR/dW in the chain rule is set to unity.

4. The authors write: ``For the eligibility trace parametrised in the polynomial form of Eq. 4, the term d eij/d theta has an explicit expression in terms of neural activations and firing rates''.
Yes, this statement is correct.  This is indeed what one can use for optimization,  but this insight  does not give equation (9) or (19). Rather it seems that the authors use this insight as a starting point for a heuristic approximation scheme.

**Ethical Concerns:**

["Major Concern: Data quality and representativeness"]

**Final Justification:**

The paper remains incomplete.  I feel that my concerns  regarding the limitations continue to be justified and I do not change my overall rating.

**Limitations:**

there are many more limitations since the theory is wrong/incomplete/inconsistent.

**Paper Formatting Concerns:**

The results section is missing (also not in supplementary).

**Quality:**

1

**Strengths And Weaknesses:**

- strengths

The aim is commendable.

- weaknesses

1. The theory sketch has many inconsistencies across equations in the main text and also in supplementary.
2. No numerical results. The main text does not contain any results. Nor does supplementary

Overall I had the impression that the authors planned to have results, but did not have time to generate them in time for submission of the main text. But the rules of NeurIPS are that all main results are summarized in the main text.

---

> ### Author Rebuttal · Authors · 2025-07-30
>
> We appreciate your time in reviewing our paper and your insightful comments. We will take them into account in the next iteration of our paper.
>
> > tau_e is the time scale of the eligibility trace. It should be around 1 second according to experimental data . In Eq. (24) the decay is replaced by a rectangular window corresponding to tau_e to infinity. This replacement is ad hoc and not mentioned in the main text. Note that with REINFORCE (as opposed to TD learning) it is close to impossible to learn tasks with rewards that are delayed longer than 2 tau_e (whereas it is simple to learn with TD!).
>
> Thank you very much for your insights. We will take them into account in the next iteration of our paper.
>
> > Eq. (6-9) in main text: I find that the detour to log-policy in Eq. (6) is not helpful (it is not even a useful analogy). Moreover Eq (7) does not lead to Eq. (8). Eq. (8) is correct, but combined with Eq. (7) it does not yield Eq. 9). Equation (19) has a square and is different from Eq. (9). Probably Eq. (9) has a mistake, because Figure 1 in Supplementary and Eq. (24) give a different equation.
>
> Yes, there are small typos in the main text. It was obviously a rushed submission.
>
> > Could the authors start with a Loss (R - bar{R})^2. ? Equation (7) is then found using chain rule if the term dR/dW in the chain rule is set to unity.
>
> This is a great idea. We will take it into our account for the next iteration of our paper.

---

> > ### Comment · Reviewer_3MaT · 2025-08-04
> > **reply to rebuttal**
> >
> > I have read the rebuttal. I will not change my rating.

---

### Official Review · Reviewer_FVJB · 2025-07-02

**Clarity:** 3
**Significance:** 2
**Originality:** 3
**Rating:** 1
**Confidence:** 4

**Summary:**

The authors introduce an approach to meta-learning synaptic update rules that can perform credit assignment via a modification to REINFORCE. The central idea is to alter the REINFORCE objective to think of the weights as implementing an effective policy. The aims introduced are to study credit assignment under this biologically realistic framework for learning rules and to determine if such rules can give rise to rich and lazy learning regimes.

**Questions:**

1. Is there a results section I somehow missed?

I will refrain from asking further questions as without a results section I cannot properly evaluate the work.

**Ethical Concerns:**

["NO or VERY MINOR ethics concerns only"]

**Final Justification:**

The authors have acknowledged their paper currently doesn't contain sufficient results for publication. Therefore, I will maintain my recommendation to reject.

**Limitations:**

Yes.

**Quality:**

1

**Strengths And Weaknesses:**

Strengths:

In my estimation, both aims introduced were very compelling. Using a REINFORCE-style loss to find plasticity suited to learn arbitrary reward functions is a nice concept and has potential. Additionally, the survey of the literature is quite thorough.

Weaknesses:

My primary concern is that there are no results. I was quite looking forward to reading that section! The referenced Extended Results section of the supplement does not appear to exist.

Also, I believe Eq. 8 is wrong. The weight updates depend linearly on the eligibility trace, as in Eq. 5, but the weights themselves should be an exponential filtration of $e_{ij} (R - \bar{R})$. One can then take the derivative with respect to $\theta$, yielding

$$ \frac{dW_{ij}}{d\theta_{kl}} = \int_0^t \delta R(t’) \frac{de_{ij}(t')}{d\theta_{kl}} dt’. $$

I believe the description given of Confavreux et al. is incorrect and actually applies to Bell et al., 2024.

Also, perhaps this approach would be improved by also considering codependent plasticity as in Agnes and Vogels, 2024. Another interesting addition might include considering multiple cell types as in Confavreux et al., 2023.

---

> ### Author Rebuttal · Authors · 2025-07-25
>
> We thank the reviewer for their thoughtful comments and for finding our work compelling.
>
>
> > My primary concern is that there are no results. I was quite looking forward to reading that section!
>
> Believe me, we were looking forward to this section more than you did, but time was not on our side.
>
> > Is there a results section I somehow missed?
>
> No, there was none in the originally submitted manuscript. We were hopping to be able to upload a one page pdf/ revised manuscript during the revision period, but unfortunately we found out this is not possible. Yet, we are grateful for the feedback you and the other reviewers provided on our work.
>
> >  Also, I believe Eq. 8 is wrong. The weight updates depend linearly on the eligibility trace, as in Eq. 5, but the weights themselves should be an exponential filtration of $e_{ij} (R - \bar{R})$.
>
> We respectfully disagree with that comment. The weight updates happen only once in each trial, at the end of the trial (discrete updates). The notation in the main text needs indeed improvement but in the supplement we explicitly highlight that the weight update depend on the eligibility trace at time T, $e^T_{ij}$.
>
> > I believe the description given of Confavreux et al. is incorrect and actually applies to Bell et al., 2024.
> Also, perhaps this approach would be improved by also considering codependent plasticity as in Agnes and Vogels, 2024. Another interesting addition might include considering multiple cell types as in Confavreux et al., 2023.
>
> We appreciate the reviewer for their insightful suggestions for extensions. Considering multiple cell types was already in our plans. More specialized plasticity rules like a firing rate analogue of the rule proposed by Agnes and Vogels, 2024 are indeed of interest for us. We appreciate your valuable input!

---

### Official Review · Reviewer_m4uo · 2025-07-02

**Clarity:** 1
**Significance:** 1
**Originality:** 1
**Rating:** 1
**Confidence:** 4

**Summary:**

This paper proposes a meta-learning algorithm that parameterizes biologically plausible plasticity rules as functions of presynaptic and postsynaptic activity and synapse size, and updates these plasticity parameters in the meta-learning outer loop. It is unclear what specific problem the paper aims to address due to the lack of experiments.

**Questions:**

- Where is the results section in the supplementary material? There is a "details on numerical implementation section" in (Appendix E), and a reference to the results section in the main paper but there is no results section.

**Ethical Concerns:**

["NO or VERY MINOR ethics concerns only"]

**Final Justification:**

Nothing to add to the original review.

**Limitations:**

Limitations are discussed about the method in general, but as no empirical evidence is given, no limitation thereof is discussed.

**Quality:**

1

**Strengths And Weaknesses:**

Weaknesses
- The paper is incomplete.
- There are references to experiments in the main text, yet there are no experiments, also not in the supplementary material
- I am unsure whether authors who do not include their code in the supplementary material can answer 'Yes' in the NeurIPS checklist.

---

> ### Author Rebuttal · Authors · 2025-07-30
>
> Thank you very much for your effort in reviewing our paper. Indeed due to time limitations we ended up not including any results in the current version of the paper. We hoped that we would be able to upload a pdf file during the revisions or an updated manuscript, but unfortunately this is not allowed any more.

---

### Decision · Program_Chairs · 2025-09-17

**Decision:**

Reject

**Comment:**

This submission is unfinished. The authors intended to provide results during the rebuttal, which is not allowed.